# Reduced miR-26b Expression in Megakaryocytes and Platelets Contributes to Elevated Level of Platelet Activation Status in Sepsis

**DOI:** 10.3390/ijms21030866

**Published:** 2020-01-29

**Authors:** Bernadett Szilágyi, Zsolt Fejes, Szilárd Póliska, Marianna Pócsi, Zsolt Czimmerer, Andreas Patsalos, Ferenc Fenyvesi, Ágnes Rusznyák, György Nagy, György Kerekes, Mariann Berhés, Ildikó Szűcs, Satya P. Kunapuli, János Kappelmayer, Béla Nagy

**Affiliations:** 1Department of Laboratory Medicine, Faculty of Medicine, University of Debrecen, H-4032 Debrecen, Hungary; sz.bernadett27@gmail.com (B.S.); fejes.zsolt@med.unideb.hu (Z.F.); pmarcsi89@gmail.com (M.P.);; 2Kálmán Laki Doctoral School of Biomedical and Clinical Sciences, Faculty of Medicine, University of Debrecen, H-4032 Debrecen, Hungary; 3Department of Biochemistry and Molecular Biology, Genomic Medicine and Bioinformatics Core Facility, Faculty of Medicine, University of Debrecen, H-4032 Debrecen, Hungary; poliska@med.unideb.hu (S.P.); czimmerer@gmail.com (Z.C.); 4Department of Medicine, Johns Hopkins University School of Medicine, Institute for Fundamental Biomedical Research, Johns Hopkins All Children’s Hospital, St. Petersburg, FL 33701, USA; apatsalos@jhmi.edu; 5Department of Pharmaceutical Technology, Faculty of Pharmacy, University of Debrecen, H-4032 Debrecen, Hungary; fenyvesi.ferenc@pharm.unideb.hu (F.F.); rusznyak.agnes@pharm.unideb.hu (Á.R.); 6Doctoral School of Pharmaceutical Sciences, University of Debrecen, H-4032 Debrecen, Hungary; 7Department of Anesthesiology and Intensive Care, Faculty of Medicine, University of Debrecen, H-4032 Debrecen, Hungary; gynagy1986@gmail.com (G.N.); bermarjan@yahoo.co.uk (M.B.); 8Institute of Internal Medicine, Faculty of Medicine, University of Debrecen, H-4032 Debrecen, Hungary; gkerekesg@gmail.com; 9Department of Pulmonology, Faculty of Medicine, University of Debrecen, H-4032 Debrecen, Hungary; ildiks@yahoo.com; 10Department of Physiology and Sol Sherry Thrombosis Center, Temple University School of Medicine, Philadelphia, PA 19140, USA; spk@temple.edu

**Keywords:** platelet activation, microRNA, SELP, megakaryocyte, sepsis, inflammation, MEG-01

## Abstract

In sepsis, platelets may become activated via toll-like receptors (TLRs), causing microvascular thrombosis. Megakaryocytes (MKs) also express these receptors; thus, severe infection may modulate thrombopoiesis. To explore the relevance of altered miRNAs in platelet activation upon sepsis, we first investigated sepsis-induced miRNA expression in platelets of septic patients. The effect of abnormal Dicer level on miRNA expression was also evaluated. miRNAs were profiled in septic vs. normal platelets using TaqMan Open Array. We validated platelet miR-26b with its target *SELP* (P-selectin) mRNA levels and correlated them with clinical outcomes. The impact of sepsis on MK transcriptome was analyzed in MEG-01 cells after lipopolysaccharide (LPS) treatment by RNA-seq. Sepsis-reduced miR-26b was further studied using Dicer1 siRNA and calpain inhibition in MEG-01 cells. Out of 390 platelet miRNAs detected, there were 121 significantly decreased, and 61 upregulated in sepsis vs. controls. Septic platelets showed attenuated miR-26b, which were associated with disease severity and mortality. *SELP* mRNA level was elevated in sepsis, especially in platelets with increased mean platelet volume, causing higher P-selectin expression. Downregulation of Dicer1 generated lower miR-26b with higher *SELP* mRNA, while calpeptin restored miR-26b in MEG-01 cells. In conclusion, decreased miR-26b in MKs and platelets contributes to an increased level of platelet activation status in sepsis.

## 1. Introduction

Sepsis is a life-threating condition with dysregulated systemic host response to microbial pathogens, while the septic shock is a subset of sepsis with circulatory, metabolic, and cellular abnormalities [1]. As a consequence, multi-organ failure may rapidly develop, resulting in early death. Altered platelet count and function are critical in this disease that contributes to sepsis-associated mortality [2]. However, platelet function in sepsis has yielded conflicting data based on previous studies. Human septic platelets have shown increased surface P-selectin expression causing high soluble P-selectin plasma levels [3], elevated thrombospondin exposure [4], and augmented platelet reactivity at early time points of the disease [5]. In contrast, hyporeactive platelets with decreased ex vivo aggregability have been reported in sepsis by others [6,7]. Whereas, an elevated level of platelet activation status in sepsis is not questioned that is associated with the upregulation of several platelet receptors [8]. P-selectin is involved in the formation of heterotypic aggregates [9], and their increased expression is associated with a higher risk for mortality, especially in older septic patients [10]. Elevated soluble P-selectin levels show a strong correlation with infection and disease severity in sepsis with coagulation disorders [11].

Based on the etiology of insults of sepsis, the pathogen-associated molecular patterns and damage-associated molecular patterns can be distinguished [1]. Among both conditions, similar mediators are released that react primarily with toll-like receptors (TLRs). Most TLR members, e.g., TLR4, are expressed on both platelets and megakaryocytes (MKs) [12]. Therefore, platelets participate in the amplified inflammatory and immune response during sepsis [13], while infection can also modulate thrombopoiesis via the TLR2 receptor [14]. Platelet hyperactivity may turn into thrombocytopenia because of neutrophil-dependent sequestration of activated platelets into the lungs in a TLR4-dependent manner [12].

Platelets carry a large number of microRNAs (miRNAs), messenger RNAs (mRNAs), and several proteins involved in miRNA processing (e.g., Dicer), which are delivered from MKs [15,16]. Expression of altered miRNAs correlates with platelet reactivity [17], while Dicer1-mediated miRNAs are able to regulate target mRNAs and de novo synthesis of proteins being important for the hemostatic function of platelets [18]. Accordingly, the expression of RNAs and related proteins are not static in platelets and can be modulated upon platelet activation [19]. Among in vitro septic conditions, mRNA levels of *IL1B* (interleukin-1β, IL-1β) and *F3* (tissue factor) are increased, and resulting proteins are translated and accumulated in human platelets in response to TLR4 agonist lipopolysaccharide (LPS) [20,21]. Based on recent in vivo experiments, transcriptional and translational properties of human and murine platelets are significantly affected by sepsis, causing de novo synthesis of αIIb protein with integrin αIIbβ3 activation [22]. However, there is no available data about how the platelet miRNA profile is altered in sepsis that, in turn, modulates target mRNA levels and platelet function.

In this study, the miRNA profile was characterized for the first time in platelets of septic patients in comparison to healthy controls. We then validated the expression of miR-26b in septic platelets with target *SELP* mRNA level that encodes P-selectin, a receptor for P-selectin glycoprotein ligand-1 [9]. This platelet miRNA was further analyzed in relation to disease severity and sepsis-related mortality. The functional relationship between miR-26b and *SELP* expression was proved among septic conditions using specific miRNA mimics. In parallel, the transcriptome of MKs in sepsis was also investigated by RNA-seq using MEG-01 cell cultures in response to LPS. Changes in the Dicer1 level were analyzed in septic platelets and LPS-induced MEG-01 cells for its contribution to altered miRNA levels. For this purpose, two experimental approaches were also applied in MEG-01 cells among septic conditions: i) downregulation of *DICER1* expression by siRNA and ii) administration of specific calpain inhibitor (calpeptin) during LPS treatment. Finally, a gene ontology (GO) analysis was performed to study the role of upregulated *SELP* expression in MK function in sepsis.

## 2. Results

### 2.1. Baseline Characteristics of Study Groups

Inflammation-dependent laboratory parameters (i.e., white blood cell (WBC) count, serum C-reactive protein (CRP), and procalcitonin (PCT)) were significantly elevated in septic subjects vs. controls (Table 1). Eighteen out of 21 subjects suffered from sepsis with pneumonia. As expected, the mean platelet count was significantly lower in the sepsis group (*p* < 0.01) than controls; however, many individual values were still within the reference range at the time of recruitment. Importantly, there was no difference in terms of administration of anti-platelet medication (e.g., aspirin, clopidogrel) between the two groups; thus, we could exclude the modulation of these regimens on platelet activation and related miRNA levels [23]. In a patient cohort, 14 individuals suffered from sepsis, while septic shock developed in seven cases, and nine subjects died of these severe clinical conditions within 28 days despite ICU treatment. In contrast, control subjects had no inflammation at the time of enrollment (Table 1).

Increased level of platelet activation was detected in septic patients based on elevated surface P-selectin expression (*p* < 0.0001), and higher plasma concentrations of soluble P-selectin (*p* < 0.0001, *n* = 10/group) in contrast to healthy controls (Figure 1A,B). In addition, increased mean platelet volume (MPV) values (*p* < 0.0001) were measured in the septic cohort vs. normal controls, reflecting an increased pool of larger and younger circulating platelets (Figure 1C), which could be more reactive than smaller counterparts [24].

### 2.2. Sepsis-Induced Changes in Platelet miRNAs

Our aim was to investigate the regulation of abnormal platelet phenotype with enhanced P-selectin expression in sepsis via measuring the level of platelet miRNAs that might be involved in platelet function. Therefore, we here profiled miRNA expression for the first time in platelets from septic patients and examined the relevance of their altered levels to elevated platelet activation status. First, a comprehensive analysis of miRNAs was performed by TaqMan Open Array in three randomly selected RNA samples of each study group to screen which platelet miRNAs were significantly altered in sepsis compared to control subjects. We considered miRNA expression significantly altered when at least 1.5-fold change was observed. Out of 754 miRNAs, 390 miRNAs could be detected in these samples: there were 121 whose levels were significantly decreased, and 61 upregulated miRNAs in septic platelets (Figure 2A). Accordingly, platelet miR-26b was found to be downregulated, which is among the 10 most abundant miRNAs in platelets [25]. In addition, miR-26b regulates P-selectin expression via regulating the *SELP* mRNA level [26]. In Appendix A, significantly altered and unchanged miRNAs of septic platelets are listed. Expression of platelet miR-26b was then validated in each enrolled septic patient, and significantly attenuated levels (*p* = 0.002) were analyzed compared to controls (Figure 2B).

To confirm the effect of septic milieu on platelet miRNA level, purified normal platelets were activated with TLR4 agonist LPS in combination with lipoprotein binding protein (LBP) and soluble CD14 at 37 °C or PBS (vehicle) for 4 h *in vitro*. For positive control, tumor necrosis factor (TNF)-α was applied to trigger platelet activation. Platelet miR-26b was significantly reduced in response to either agonist (Appendix A). These results were in agreement with data of ex vivo septic platelets, suggesting that LPS acted as a direct platelet activator and could alter RNA expression in sepsis, similar to former findings [20].

### 2.3. Platelet miR-26b Expression Correlates with Sepsis Severity and Mortality

We next investigated whether changes in platelet miR-26b expression were associated with the severity of sepsis and the clinical outcome in the presence of elevated platelet P-selectin positivity. Patients were separately analyzed based on the progression of their disease, i.e., sepsis (*n* = 14) or septic shock (*n* = 7). Significantly lower levels of miR-26b were found in patients who had septic shock than sepsis (*p* = 0.036) (Figure 2C). When we sub-grouped septic individuals into ‘survivors’ (*n* = 9) and ‘non-survivors’ (*n* = 12) cohorts according to 28-day mortality, even lower platelet miR-26b expression (*p* = 0.022) was measured in early death compared to surviving patients (Figure 2D). These results suggested that there was a remarkable alteration in the expression of platelet miR-26b in connection with the severity and mortality of sepsis.

### 2.4. SELP Expression is Upregulated in Platelets with Sepsis

We then asked whether changes in the expression of target mRNA of miR-26b could be also observed in these septic platelets; thus, the expression of *SELP* was quantified by RT-qPCR. As the level of *IL1B* mRNA was earlier revealed to be processed and accumulated in sepsis-induced platelets [20], that is why *IL1B* expression was parallelly studied that was substantially induced (*p* = 0.002) in the ex vivo sepsis samples (Figure 3A). Importantly, *SELP* mRNA targeted by miR-26b [26] was upregulated in sepsis (*p* < 0.001) relative to healthy platelets (Figure 3B). Furthermore, higher *SELP* and *IL1B* mRNA levels were observed in the case of in vitro stimulated normal platelets by LPS (Appendix A). Accordingly, TLR4 signaling dependent platelet activation events were associated with induced *SELP* expression. Platelet *SELP* mRNA levels were higher in subjects who suffered from septic shock (Figure 3C) or did not survive (Figure 3D), but these differences did not reach a statistical significance probably due to the limited number of patients. When we compared *SELP* expression based on the values of MPV, significantly higher *SELP* mRNA levels (*p* = 0.008) were found in those showing larger than normal MPV values (≥11.1 fL) (Figure 3E).

Due to the increased *SELP* expression in patient platelets after the development of sepsis, we wondered whether P-selectin at protein level could be also detected at a higher quantity in platelets. We found that platelet lysates obtained after 72 h of the onset of sepsis showed a significantly larger amount of P-selectin compared to control specimens using Western blotting (Figure 3F). These results suggested that more P-selectin was available in larger (i.e., newly synthesized) platelets that could provide excessive functional responses with higher P-selectin expression in sepsis [3,8].

### 2.5. TLR4 Modulated Transcriptome in MEG-01 Cells Via Nuclear Factor-Kappa B (NF-κB) Pathway

In order to confirm the cellular activation of MEG-01 cells by LPS via the nuclear factor-kappa B (NF-κB) pathway, first, the nuclear translocation of p65 was visualized by fluorescence microscopy. As a positive control, TNF-α treatment was used in the same settings. There was an enhanced nucleus/cytosol intensity for p65 after LPS or TNF-α vs. baseline (PBS) samples (Figure 4A,B).

To explore the functional impact of altered RNA expression at the level of MK in sepsis, we thoroughly investigated LPS-induced transcriptional changes in MEG-01 cells. We stimulated MEG-01 cell cultures with LPS for 4 h and determined their transcriptome by RNA-seq. Transcriptomic and differential expression analysis by RNA-seq revealed 1414 differentially expressed (DE) genes (Figure 5A). More specifically, we identified 354 significantly upregulated and 1060 significantly downregulated transcripts in LPS-activated MEG-01 cells at a fold change of ≥1.5 compared to untreated cells (Figure 5A, Appendix A). A clustered heatmap with the top 50 differentially expressed genes is shown in Figure 5B. These results revealed that there might be substantial alterations in the MK transcriptome during sepsis.

To assess the biological response and functions that TLR4-activated MKs could gain at the level of gene expression, we extended our study with pathway analysis of the altered genes in MEG-01 cells after 4-h LPS stimulation, using the gene ontology resource database (Appendix A). Interestingly, LPS treatment significantly affected genes associated with the activation of several inflammatory pathways involving cell responses to TNF-α, IL-1β, interferon-gamma, and LPS, as well as gene changes associated with leukocyte migration and chemotaxis (Appendix A), all indicative of a potential acquisition of a pro-inflammatory-like phenotype. Other notable pathways affected in MKs in response to TLR4 agonist included genes changes associated with response to calcium, lipid homeostasis, angiogenesis, and the MAPK, ERK1/2, and JNK signaling pathways. Importantly, *SELP*, which was significantly upregulated in MKs in response to LPS, could be implicated in several different predicted pathways, highlighted in Appendix A, including the ones in response to LPS and inflammation, leukocyte adhesion and migration, as well as implications with sequestering calcium ions. Hence, the TLR4-dependent pathway could substantially modulate gene expression in MKs [27] and might act as a potential link between the inflammatory environment and altered gene expression profile of this cell type. These findings also supported that *SELP*, encoding a key cellular adhesion molecule P-selectin, had a central role in the regulation of inflammatory signaling of MKs upon sepsis. To verify the reproducibility of our RNA-seq analysis, we validated the expression of *SELP* in LPS-stimulated MKs that showed a significant rise (Figure 5C).

### 2.6. Sepsis-Reduced miR-26b Results in Upregulated SELP Expression

Similar to the findings in septic platelets, miR-26b was significantly downregulated in MEG-01 cells in response to LPS (Figure 5D). These findings suggested that sepsis could trigger MKs to alter miRNA levels as well. Although there were some former pieces of evidence that miR-26b regulates *SELP* expression in hyperglycemia [26], we wanted to confirm the functional relationship between miR-26b and *SELP* mRNA under inflammatory conditions. As LPS resulted in a lowered miR-26 level in MEG-01 cells, we here applied specific miRNA mimics to artificially overexpress this miRNA in order to investigate the change in *SELP* expression (Figure 5E). We found that highly elevated miR-26b expression reduced the level of SELP mRNA in MEG-01 cells, with about 50% compared to those samples treated with NEG-01 mimic (Figure 5F). Of note, this manipulation did not alter the expression of other miRNAs, such as miR-223 (not shown). These data underlined the fact that miR-26b targeted *SELP*, and decreased miR-26b contributed to elevated *SELP* expression of MKs and platelets in sepsis.

### 2.7. Decreased Dicer1 Level in Platelets and MKs Results in Abnormal miRNA Among Septic Conditions

We next focused on the mechanism via mature miRNAs that were dysregulated in sepsis. Alteration of the Dicer1 level was first analyzed by Western blotting in septic vs. control platelets. There was a significantly lower Dicer1 protein level in platelet lysates of septic individuals (Figure 6A). Similarly, LPS-stimulated MEG-01 cells showed decreased protein expression of Dicer1 than control samples detected by fluorescence microscopy (Figure 6B,C).

We then applied two additional approaches to further investigate the direct effect of abnormal Dicer level on miRNA in sepsis: i) by silencing of *DICER1* expression using siRNA to mimic sepsis-reduced Dicer function in MEG-01 cells and ii) via specific inhibition of calpain 1 and calpain 2 with calpeptin to modulate the level of miR-26b in these cells. For this purpose, we transfected MEG-01 cells with Dicer1 specific siRNA for 24 h to downregulate *DICER1* expression. The efficacy of transfection was confirmed via the quantification of the Dicer1 siRNA level, showing highly elevated expression (Figure 7A). This manipulation resulted in lowered *DICER1* mRNA vs. control samples with NEG-01 siRNA (Figure 7B). As a consequence, miR-26b was significantly decreased (*p* < 0.05) (Figure 7C), while *SELP* mRNA level was upregulated (*p* < 0.05) (Figure 7E). In parallel, we also analyzed the expression of Dicer1-independent miR-451 to check the specificity of this intervention, and no change was seen in this miRNA (Figure 7D). To test the role of lower Dicer activity on decreased platelet miRNAs, additional experiments were performed when calpeptin inhibited the calpain function induced by increased intracellular Ca^2+^ concentration upon LPS or TNF-α-triggered signaling [28]. When calpain function was blocked, and the cleavage of Dicer was prevented by calpeptin in the presence of LPS (Figure 7F) or TNF-α (data not shown), the mature miR-26b level was restored in MEG-01 cells. These data supported our hypothesis that abnormal Dicer activity in sepsis could reduce miR-26b in both platelets and MKs.

## 3. Discussion

TLR4 receptor is functional on both platelets and MKs [12]. In sepsis, platelets can be stimulated by LPS via TLR4, resulting in primed platelet activation elicited by other agonists [29,30]. Hence, there is a key role of the TLR4 receptor in the modulation of platelet phenotype in sepsis [13,27]. On the other hand, severe inflammation via TLR2 also regulates MK function that affects platelet production and function with enhanced GPIb and COX-2 protein expression [14]; however, no data has been published about TLR4 in this context. In former animal models, when mice were exposed to a sublethal/low dose of LPS for up to 1 week, platelets became gradually activated, showing high P-selectin surface positivity and a larger sensitivity to aggregation as a reflection to the action of LPS on MKs [31,32]. Very recently, *ITGA2B* (integrin subunit αIIb) expression has been found to be upregulated in circulating platelets during sepsis via dynamic trafficking of specific mRNA from MKs, and this was accompanied by increased production of integrin subunit αIIb and activation of integrin αIIbβ3 [22]. All these changes in MKs may provide a pro-thrombotic phenotype of platelets in sepsis that could affect the procoagulant activity of the blood with increased risk for thrombosis [27]. Of note, there was no former data on how platelet P-selectin expression can be regulated via the MK-platelet axis during sepsis.

Recently, the potential role of miRNAs in platelet and MK function has also got in focus [25,33]. For instance, platelet miR-27b can regulate platelet synthesis of trombospondin-1 [19], or miR-15a modulates GPVI-mediated αIIbβ3 activation and α-granule release in MKs [34]. In sepsis, there was only one former publication, which reported impaired miRNA levels in exosomes and pooled human blood cells but not in platelets [35]. Since leukocytes and platelets consist of the different repertoire of miRNAs [36], here we sought to investigate the miRNA profile of purified platelets of septic patients.

First, we profiled miRNA expression in randomly selected platelet samples from three septic patients by TaqMan Open Array. In comparison to normal individuals, 121 downregulated and 61 upregulated miRNAs were found in the septic platelets vs. controls. In the entire sepsis group consisting of 21 patients showing an increased level of platelet activation, we validated the expression of platelet miR-26b that regulates *SELP* expression [26]. This platelet miRNA showed significantly reduced levels than normal. Highly attenuated levels of platelet miR-26b were associated with the development of septic shock and early death. Accordingly, this platelet miRNA might act as a reliable biomarker for indicating platelet reactivity in this disease, as miRNAs have been recently suggested for such clinical reasons [37,38]. Similar to intracellular miRNA expression, reduced levels of its circulating form have been effectively used as laboratory biomarkers in serum samples of septic shock subjects [35]. In addition, the level of miR-199b in peripheral blood cells correlates with disease severity, while exosomal miR-125b predicts survival in sepsis [35].

Septic platelets demonstrated augmented *SELP* mRNA level compared to controls in the presence of elevated *IL1B* expression that was earlier described in septic platelets [20]. Importantly, when we further analyzed *SELP* expression based on the values of MPV, higher *SELP* mRNA levels were found in those with platelets having larger MPV values (≥ 11.1 fL). There is a large number of papers reporting MPV as a measure of platelet size and activity as well [24]. Higher MPV has been associated with various disease conditions, for example, in patients with acute coronary syndrome [24]. Larger platelets usually contain more secretory granules, more RNA, and thus become more reactive than their smaller counterparts [39]. Although *SELP* expression did not significantly correlate with the outcome of sepsis in these patients, increased MPV values also predicted disease prognosis, as reported by others in sepsis with pneumonia [40]. In the presence of an augmented mRNA level that predicts changes in protein expression, higher P-selectin concentration was detected in platelet lysates after 72 h of sepsis onset that could be a result of altered miRNA and mRNA levels in platelets and MKs. Based on a recent animal model with peritoneal sepsis, mice showed increased P-selectin positivity at 48 h as a part of the prothrombotic phenotype of platelets developed in sepsis [41]. Increased P-selectin expression on activated platelets is highly involved in the formation of heterotypic aggregates, resulting in microvascular thrombosis [9]. The blockade of platelet P-selectin, in combination with simultaneous inhibition of CD11b receptor, on neutrophils effectively attenuates platelet-neutrophil interactions in septic shock [42]. Thus, enhanced *SELP* expression in platelets might contribute to a higher risk for cellular interactions and might represent a new therapeutic target in those with sepsis.

We started from some evidence that MKs are also affected by sepsis that results in altered mRNA levels in circulating platelets [22,43]. Hence, we thoroughly investigated LPS-induced transcriptional changes in MEG-01 cell cultures since increased *ITGA2B* mRNA content has been observed in MKs to be invested in platelets during sepsis [22]. For this purpose, we stimulated MEG-01 cell cultures with LPS for 4 h to analyze the transcriptome of MKs. LPS could induce the activation of the NF-κB pathway in MEG-01 cells, as earlier observed via TLR2 signaling [14], that we visualized with the enhanced nuclear translocation of p65 by fluorescence microscopy. Using RNA-seq, 1060 significantly downregulated and 354 upregulated transcripts were detected in LPS-activated MKs. Based on this analysis, *SELP* was identified among the top 25 most upregulated genes. We then validated the expression of *SELP* in LPS-stimulated MEG-01 cells, showing a significant elevation. To compare our data with recently published results in sepsis [22], the expression of *ITGA2B* was also analyzed in our septic platelets and LPS-activated MK cell cultures by RT-qPCR, which were upregulated vs. controls (not shown). These results revealed that there must occur substantial alterations in the MK transcriptome after the onset of sepsis. Modulated RNA expression might result in transcriptional and translational events via the trafficking of RNA content through the MK-platelet axis that was found in the background of increased integrin subunit αIIb and granzyme B production in septic platelets [22,43].

In LPS-stimulated MEG-01 cell samples, miR-26b was downregulated, similar to ex vivo septic platelets. In type 2 diabetes mellitus, decreased platelet miR-26b has been associated with elevated *SELP* expression, resulting in higher platelet reactivity [26]. In contrast, after cardiopulmonary bypass, overexpression of platelet miR-10b and miR-96 decreases mRNAs of GPIb and VAMP8, as well as their protein levels, causing defected platelet function [44]. The modulatory effect of miR-26b on *SELP* expression was confirmed in MEG-01 cells under these inflammatory conditions using a specific miRNA mimic, suggesting that sepsis-reduced miR-26b caused increased *SELP* expression in sepsis.

Abnormal Dicer1 activity has been found to be an important factor in dysregulated miRNAs that have been revealed in Dicer1-deficient murine platelets [18] and in diabetic platelets, showing decreased Dicer function [26,45]. Dicer enzyme is a substrate of calpain 1 (μ-calpain) and calpain 2 (m-calpain), which can be found in platelets [45]. In sepsis, there was no data about how the Dicer function modulates miRNA expression. We here observed decreased Dicer1 levels in platelet lysates of septic individuals. Similarly, LPS-stimulated MEG-01 cells showed decreased Dicer1 expression. Direct investigation of Dicer1 function with its gene silencing by siRNA and through specific inhibition of calpain 1 and calpain 2 with calpeptin to modulate miRNAs in MEG-01 cells revealed that abnormal Dicer1 activity was generated in sepsis that could reduce miR-26b and, in turn, to elevate *SELP* expression in both platelets and MKs. Accordingly, elevated intracellular Ca^2+^ concentration in response to LPS or TNF-α induces calpain function that cleaves the Dicer enzyme, causing less mature miRNAs [28]. When the function of calpain 1 and 2 was blocked, and the cleavage of Dicer was prevented by calpeptin in the presence of any of these inflammatory mediators, mature miR-26b levels were restored in MEG-01 cells. Accordingly, changes in Dicer activity due to sepsis might occur in both platelets and MKs, shaping the profile of miRNAs. Based on these current results, we proposed a signaling axis in MKs and platelets upon sepsis when lower Dicer level resulted in decreased miR-26b with elevated target *SELP* expression that could contribute to the elevated level of platelet activation status.

Finally, we extended our study with a GO analysis. Upregulated *SELP* seemed to be involved in seven different pathways of MKs in TLR4 involvement, such as in the regulation of inflammatory response. High *SELP* expression in peripheral blood cells has been investigated as a new risk factor in rheumatoid arthritis [46], and some specific haplotypes of *SELP* gene have been related to a higher risk for myocardial infarction [47], but no former data of altered *SELP* expression were available in regard to platelet/MK function in sepsis. According to these current data, *SELP* might play a central role via inflammatory signaling of MKs by LPS apart from encoding a key cellular adhesion molecule P-selectin.

There were some limitations to this study. First, a limited number of septic patients could be involved in this study due to the strict criteria of enrollment. Second, we did not investigate the role of other LPS-independent mechanisms of altered platelet miRNA expression upon sepsis. Hence, further studies are required to examine the details of this complex mechanism.

In conclusion, septic platelets showed an altered miRNA profile with 182 abnormally expressed miRNAs via TLR4. Reduced platelet miR-26b correlated with sepsis severity and mortality; thus, it might become a useful biomarker for indicating elevated platelet activation status in this disease. Upregulated *SELP* expression in MKs through TLR4 resulted in enhanced P-selectin expression in platelets and might be also involved in shaping inflammatory responses of MKs.

## 4. Materials and Methods

### 4.1. Study Design, Participants, and Blood Sample Preparation

In this analyst-blinded, case-control study 21 patients with a primary diagnosis of sepsis (16 males, 5 females, aged 64 (51–70) years) were prospectively included into this study within 24 h of admission to the intensive care unit (ICU) of one of the three clinical departments (Table 1). Sepsis was diagnosed based on the criteria of the American College of Chest Physicians/Society of Critical Care Medicine Consensus, which defined systemic infection and 2 of the following: (a) temperature >38 °C or <36 °C; (b) heart rate >90 beats/min; (c) respiratory rate >20 breaths/min or PaCO_2_ <32 mm Hg; (d) WBC count >12,000/mm^3^, <4000/mm^3^, or >10% bands [48]. The sequential organ failure assessment (SOFA) score was determined by the clinicians, and the administration of any antiplatelet agents was recorded in each case. Exclusion criteria for enrollment included malignancy, autoimmune disease, pregnancy, severe thrombocytopenia, and acute myocardial infarction or acute ischemic stroke within 1 month. All-cause 28-day mortality was recorded prospectively.

To investigate platelet miRNAs and mRNA levels in sepsis, venous blood samples were obtained from patients by atraumatic venipuncture into Vacutainer^®^ tubes containing 0.105 M sodium citrate (Becton Dickinson, San Jose, CA, USA). Samples were prepared within 60 min after sampling and were centrifuged at 170 × *g* for 15 min at room temperature (RT) to obtain platelet-rich plasma (PRP). The upper layer of PRP was carefully transferred to a plastic tube to avoid any leukocyte contamination. In the case of 7 septic patients, follow-up samples were also obtained after 72 h of sepsis onset.

In parallel, 21 age- and gender-matched controls (14 males, 7 females, aged 58 (42–65) years) were enrolled among volunteers or staff members from the Departments of Laboratory Medicine and Internal Medicine who underwent a detailed medical history, physical examination, and routine laboratory tests and were free of acute cardiovascular, metabolic, inflammatory diseases, or cancer (Table 1). All participants gave written informed consent. The study was approved by the Ethics Committee of the University of Debrecen (permit number: 4780-2017) in accordance with the Declaration of Helsinki.

### 4.2. Leukocyte-Depleted Platelet Preparation

Leukocyte-depleted platelet samples (LDP) were purified by anti-CD45-conjugated magnetic microbeads (Dynabeads^®^, Invitrogen, Oslo, Norway) within 30 min of blood sampling, as we previously described from our laboratory [26]. Briefly, after the incubation of 2 mL PRP with the beads for 30 min at RT, samples were inserted into a magnetic separator (Becton Dickinson, San Jose, CA, USA) for 2 × 2 min, and LDP was then transferred into a fresh tube for additional centrifugation (1500× *g*, 15 min, RT). Platelet pellet was lysed with 750 μL TRI reagent (Molecular Research Center Inc, Cincinnati, OH, USA) and stored at −20 °C before RNA isolation.

### 4.3. Total RNA Extraction

Total RNA from LDP and MK cell culture samples was isolated by TRI reagent according to the manufacturer’s recommendations. The purity and the concentration of separated RNA samples were verified by a NanoDrop 2000 spectrophotometer (Thermo Scientific, Wilmington, DE, USA). Total RNA samples were stored at −80 °C.

### 4.4. TaqMan Open Array-Based miRNA Profiling in Septic Platelets

First, we randomly selected 3 total RNA samples from the septic and control groups, and we analyzed 754 types of miRNA using a TaqMan Open Array technology (Applied Biosystems, Foster City, CA, USA) following the manufacturer’s protocol. Briefly, 100 ng of total RNA was used for reverse transcription with TaqMan MicroRNA Reverse Transcription Kit (Applied Biosystems, Foster City, CA, USA) and Megaplex RT Primers Human Pool Set v3.0 (Applied Biosystems, Foster City, CA, USA). The reactions were performed for 40 cycles of 16 °C for 2 min, 42 °C for 1 min, 50 °C for 1 s, and 1 cycle of 85 °C for 5 min. Then, specific complementary DNA (cDNA) samples were pre-amplified with Megaplex PreAmp Primers Human Pool Set v3.0, as well as TaqMan PreAmp Master Mix (Applied Biosystems, Foster City, CA, USA), to increase the quantity of the desired cDNA. PCR reactions were run at the following conditions: 95 °C for 10 min, 55 °C for 2 min, 72 °C for 2 min, and 12 cycles of 95 °C for 15 s and 60 °C for 4 min. Diluted pre-amplification products (1:40) and PCR reaction mix containing TaqMan Open Array Real-Time PCR Master Mix (Applied Biosystems, Foster City, CA, USA) were transferred into a 384-well plate, and the Open Array AccuFill system loaded the samples to the prepared TaqMan Open Array Human MicroRNA panels (Applied Biosystems, Foster City, CA, USA). Finally, plates were run on a QuantStudio 12 K Flex qPCR instrument (Applied Biosystems, Foster City, CA, USA). For data normalization, the RNU-48 control assay was used in this experiment. Data were analyzed with Thermo Fisher Cloud System (Thermo Fischer Scientific, Waltham, MA USA) and Expression Suite Software v1.0.3 (Applied Biosystems, Foster City, CA, USA).

### 4.5. miRNA Specific Stem-Loop RT-qPCR Analysis

The expression of selected platelet miRNAs in the entire study groups was validated by miRNA specific Universal ProbeLibrary (UPL)-probe based stem-loop RT-qPCR method [26]. Briefly, this quantification technique included two steps: (1) miRNAs (input total RNA: 10 ng) were transcribed into cDNA via reverse transcription using miRNA-specific stem loop-RT primer (500 nM, Integrated DNA Technologies, Leuven, Belgium) and TaqMan^®^ MicroRNA Reverse Transcription Kit (Applied Biosystems, Foster City, CA, USA) and (2) miRNA quantification was performed by RT-qPCR using designed universal reverse primer (100 μM, Sigma-Aldrich, St. Louis, MO, USA), miRNA-specific forward primer (100 μM, Integrated DNA Technologies, Leuven, Belgium), and UPL probe #21 (10 μM, Roche Diagnostics, Mannheim, Germany) with Taq polymerase (5 U/μL, Thermo Scientific, Wilmington, DE, USA) and dNTPs (2.5 mM, Thermo Scientific). The reactions were pre-incubated at 95 °C for 1 min, followed by 40 cycles of 95 °C for 15 s and 60 °C for 30 s. All the measurements were conducted in triplicate on a QuantStudio 12 K Flex qPCR instrument (Applied Biosystems, Foster City, CA, USA). For normalization, the small-nucleolar RNU-43 was measured as a reference gene, similarly used in our former study [26]. Primers and qPCR assays were designed by the software developed by Czimmerer et al. [49], and oligonucleotides that were used in this study are listed in Appendix A.

### 4.6. mRNA Specific RT-qPCR Analysis

Complementary DNA (cDNA) synthesis was performed with a High Capacity cDNA Reverse Transcription kit (Applied Biosystems, Foster City, CA, USA) according to the manufacturer’s recommendation. The initial amount of RNA in LDP was 200 ng per reaction, while 1000 ng per reaction was used in the MK experiments. Quantitative PCR was performed on a QuantStudio 12 K Flex qPCR instrument (Applied Biosystems, Foster City, CA, USA) with Light Cycler 480 SYBR Green I Master mix (Roche Diagnostics, Mannheim, Germany) and gene-specific primers (10 μM, Integrated DNA Technologies, Leuven, Belgium). The reactions were incubated at 95 °C for 10 min, followed by 40 cycles of 95 °C for 10 sec and 60 °C for 1 min. For normalization, we used the reference gene RPLP0 (36B4). Sequences of the primers for mRNAs are also listed in Appendix A.

### 4.7. In Vitro Activation of Normal Human Platelets by LPS

LDP samples prepared from specimens of 5 healthy volunteers were treated with vehicle (PBS) or LPS (O55:B5, 100 ng/mL, Sigma-Aldrich, St. Louis, MO, USA) in the presence of lipoprotein binding protein (LBP, 100 ng/mL, Sigma-Aldrich, St. Louis, MO, USA) and soluble CD14 (150 ng/mL, Sigma-Aldrich, St. Louis, MO, USA) at 37 °C for 4 h, as formerly described by others [20]. For positive control, tumor necrosis factor-α (TNF-α) (100 ng/mL, Gibco, Grand Island, NY, USA) was used as a key pro-inflammatory mediator expressed upon TLR4 activation. Platelets were then centrifuged (1500 g, 15 min, RT), and platelet pellet was lysed with 750 μL TRI reagent and stored at −20 °C before RNA isolation. To evidence the activation of the TLR4 pathway in platelets by LPS, *IL1B* mRNA level was quantified in parallel to *SELP* expression by RT-qPCR.

### 4.8. Culturing of MEG-01 Cells Mimicking Septic Conditions

Human megakaryoblastic leukemia cell line MEG-01 cells (Sigma-Aldrich, St. Louis, MO, USA) were cultured in RPMI-1640 medium (Sigma-Aldrich, St. Louis, MO, USA) with 10% fetal bovine serum (FBS, Sigma-Aldrich, St. Louis, MO, USA), 100 U/mL Penicillin, and 100 μg/mL Streptomycin (Sigma-Aldrich, St. Louis, MO, USA) at 37 °C, 5% CO_2_. The cell count was set to 0.3 × 10^6^/mL, similar to our recent study [26]. MKs were stimulated with LPS (O55:B5, 100 ng/mL, Sigma-Aldrich, St. Louis, MO, USA) in the presence of LBP (100 ng/mL, Sigma-Aldrich, St. Louis, MO, USA) and soluble CD14 (150 ng/mL, Sigma-Aldrich, St. Louis, MO, USA) for 4–24 h to maintain them under in vitro ‘septic’ conditions as applied for the analysis of sepsis-induced platelet activation [20]. In positive control samples, MEG-01 cells were treated with TNF-α (100 ng/mL, Gibco, Grand Island, NY, USA), while negative control samples were cultured with vehicle (PBS). After treatment, cells were washed once with sterile PBS, then lysed in 750 μL TRI reagent, and stored at −20 °C before RNA isolation. To demonstrate the inflammation specific activation of MEG-01 cells via the NF-κB pathway, *IL1B* expression was analyzed by RT-qPCR.

### 4.9. RNA-Sequencing

To obtain global transcriptome data of LPS-stimulated MKs, high throughput mRNA sequencing analysis was performed on Illumina Sequencing Platform (Illumina, San Diego, CA, USA). For this purpose, 3 sets of MEG-01 cells (0.3 × 10^6^ cell/mL) were cultured in the presence of LPS or vehicle for 4 h, as described above. Total RNA was extracted and quantified, and RNA sample quality was checked on Agilent BioAnalyzer using Eukaryotic Total RNA Nano Kit (Agilent Technologies, Santa Clara, CA, USA) according to the manufacturer’s protocol. Samples with RNA integrity number (RIN) value > 7 were accepted for the library preparation process.

RNA-Seq libraries were prepared from total RNA (200 ng) using NEBNext^®^ Ultra II RNA Sample Preparation Kit for Illumina (New England BioLabs, Ipswich, MA, USA) according to the manufacturer’s protocol. Briefly, poly-A tailed RNAs were purified by oligodT-conjugated magnetic beads and fragmented at 94 °C for 15 min. First-strand cDNA was generated by random priming reverse transcription, and the second strand synthesis step was performed to generate double-stranded cDNA. After the repairing ends and adapter ligation steps, adapter-ligated fragments were amplified in enrichment PCR, and, finally, sequencing libraries were generated. The sequencing run was executed on Illumina NextSeq500 instrument (Illumina) using single-end 75 cycles sequencing. Aligned sequencing data were deposited into the NCBI SRA database under accession no. PRJNA587604.

### 4.10. Analysis of RNA-Seq Data

Raw sequencing data (fastq) was aligned to the human reference genome version GRCh37 using the HISAT2 algorithm, and BAM files were generated. Downstream analysis was performed using StrandNGS software (www.strand-ngs.com). BAM files were imported into the software, and the DESeq1 algorithm was used for normalization. To identify differentially expressed genes between untreated and LPS-stimulated conditions, ANOVA with Tukey *post hoc* test was used. Heatmaps and dot plots were drawn using R packages *pheatmap* and *ggplot2*.

### 4.11. Gene Ontology Analysis

Lists of differentially expressed genes were analyzed using the Panther tool (http://www.geneontology.org/) and the GO enrichment analysis function to create a GO. GOs with fold enrichment ≥ 2 and *p*-value < 0.05 were selected, and results were presented according to their -log_10_
*p*-value. The bar graph was drawn using the R package *ggplot2.*

### 4.12. Transfection of MEG-01 Cells with miR-26b Mimic

MEG-01 cells pretreated with LPS or TNF-α (100 ng/mL) for 4 h were centrifuged and resuspended in Opti-MEM I Reduced Serum Medium (Gibco, Grand Island, NY, USA) with 3% FBS, 100 U/mL Penicillin, and 100 μg/mL Streptomycin. The overexpression of miRNAs was performed using mirVana^®^ miR-26b mimic (40 pmol, Ambion, Austin, TX, USA) with Lipofectamine RNAiMAX^®^ Transfection Reagent (Invitrogen, Carlsbad, CA, USA) for 24 h at 37 °C and 5% CO_2_. In parallel, the negative control sample was treated with mirVana^®^ miRNA mimic negative control (NEG-01, 40 pmol, Ambion, Austin, TX, USA). After transfection, total RNA was extracted, miR-26b expression was quantified with *SELP* mRNA, as described above.

### 4.13. Western Blot

Isolated platelets obtained from septic and control individuals were lysed in RIPA buffer containing a protease inhibitor mix (Sigma-Aldrich, St. Louis, MO, USA). Proteins were separated by electrophoresis using 7.5% polyacrylamide gel and then transferred onto a nitrocellulose membrane (Bio-Rad, Hercules, CA, USA). After blocking with Tris-buffered saline/Tween (TBST; 20 mM Tris, 140 mM NaCl, 0.1% (vol/vol) Tween 20) containing 5% bovine serum albumin (BSA, Sigma-Aldrich, St. Louis, MO, USA) for 90 min at RT, membranes were incubated with monoclonal mouse anti-human Dicer1 (ab14601, 1:100, Abcam, Cambridge, UK) or monoclonal mouse anti-human P-selectin (sc-19672, 1:100, Santa Cruz Biotechnology, Dallas, TX, USA) antibody in TBST with 5% BSA at 4 °C for overnight with gentle agitation, respectively. Anti-β-actin antibody (ab8227, 1:1000, Abcam, Cambridge, UK) was used to ensure equal protein concentrations in all lanes. Membranes were labeled with HRP-conjugated goat anti-mouse secondary antibody (1:100,000, Bio-Rad, Hercules, CA, USA) for 1 h at RT, and immunoreactivity was visualized by Immobilon Western Chemiluminescent HRP Substrate (Millipore, Billerica, MA, USA). The relative intensity of P-selectin or Dicer1 bands was determined in both septic and control samples by normalization to β-actin.

### 4.14. Flow Cytometry

Investigation of platelet activation level via surface P-selectin expression on platelets was performed, as previously reported [26]. Briefly, 40 μL of whole blood samples were fixed in 1 mL 1% PFA and kept at RT for 1 h. Platelets were identified by a FITC-conjugated monoclonal antibody to GPIX (CD42a-FITC, Becton Dickinson, San Jose, CA, USA). Platelet activation was detected by phycoerythrin (PE)-labeled anti-P-selectin (CD62-PE, Becton Dickinson, San Jose, CA, USA). Fixed platelets were incubated with saturating concentrations of antibodies for 20 min in the dark at RT. As a control for immunolabeling with anti-CD62 antibody, platelets were incubated with PE-coupled non-immune mouse IgG_1_ antibody (Becton Dickinson, San Jose, CA, USA). A total of 10,000 dual-color labeled platelet events were acquired on an FC-500 flow cytometer (Beckman Coulter, Pasadena, CA, USA). Results were expressed as the percentage of double-positive platelets.

### 4.15. Fluorescence Microscopy

Detection of NF-κB activation in MEG-01 cells was visualized via p65 nuclear staining based on our previous study [50] with some minor modifications. MEG-01 cells were cultured on 6-well plates for 2 days, were then treated with LPS or vehicle (PBS) for 4 h, and were fixed with ice-cold methanol-acetone (50 *v*/*v*%) for 10 min. These cells were transferred onto sterile uncoated microscope slides at a density of 5 × 10^4^ cells/slide. Non-specific antibody binding sites were blocked with FBS (Sigma-Aldrich, St. Louis, MO, USA) for 15 min. For primary labeling of NF-κB p65 subunit, polyclonal rabbit anti-human p65 antibody (100 μg/mL, Sigma-Aldrich, St. Louis, MO, USA) was used for 1 h followed by secondary staining with Alexa Fluor 488-conjugated goat-anti-rabbit IgG (5 µg/mL, Sigma-Aldrich, St. Louis, MO, USA) for 1 h.

The protein level of Dicer1 in MEG-01 cells was also studied by fluorescence microscopy. MEG-01 cells were treated with LPS or PBS for 24 h, and fixed cells were stained by mouse anti-human Dicer1 antibody (2 µg/mL, Abcam, Cambridge, UK) followed by secondary staining with Alexa Fluor 488-conjugated goat anti-mouse IgG (5 µg/mL, Invitrogen, Carlsbad, CA, USA).

During both analyses, cell nuclei were labeled with Hoechst 33,342 (Invitrogen, Carlsbad, CA, USA), and samples were observed by Zeiss Axio Scope.A1 fluorescent microscope (HBO 100 lamp) (Carl Zeiss Microimaging GmbH, Goettingen, Germany). DAPI: excitation 365 nm, emission BP 445/50 nm; fluorescein: excitation BP 470/40 nm, emission BP525/50 nm. Images were analyzed with ZEN 2012 v.1.1.0.0. software (Carl Zeiss Microscopy GmbH, Goettingen, Germany). The ratio of nuclear and perinuclear fluorescence intensity was calculated for NF-κB p65 staining, while fluorescence intensity in the cytoplasm was determined for Dicer1 positivity. The specificity of immunostaining was checked by incubating the cells with the secondary antibody only, and no background staining was found.

### 4.16. Downregulation of DICER1 Expression by siRNA Transfection in MEG-01 Cells

To investigate whether abnormal Dicer1 function could markedly affect miR-26b levels of MKs in sepsis, *DICER1* expression was first silenced by specific siRNA (40–80 pmoL, ID: S23756, Invitrogen, Carlsbad, CA, USA) in MEG-01 cells (0.3 × 10^6^/mL) for 24–48 h in comparison to control samples with NEG-01 siRNA (Silencer Select Negative control No.1, Invitrogen, Carlsbad, CA, USA) according to the manufacturer’s recommendations. Total RNA was then isolated, and the efficacy of transfection was monitored via the quantification of the Dicer1 siRNA level by TaqMan siRNA assay (ID: S23756_asy, Invitrogen, Carlsbad, CA, USA). Expressions of miR-26b with Dicer-independent miR-451 and *DICER1* with *SELP* mRNAs were subsequently measured by RT-qPCR after transfection.

### 4.17. Analysis of Dicer Function on miRNA Level Through Calpain Inhibition in MEG-01 Cells Among Inflammatory Conditions

To examine the contribution of calpain substrate Dicer enzyme in the generation of altered miRNA levels in sepsis, we applied a MK model for sepsis-induced Dicer dysfunction in which 0.3 × 10^6^/mL MEG-01 cells were treated by a specific exogenous calpain 1 and 2 inhibitor, calpeptin (40 μmol/L, Sigma-Aldrich, St. Louis, MO, USA) for 24 h in a similar way as it was performed previously [21,40]. The effect of calpeptin on miRNA expression was assessed in the following settings: vehicle (DMSO); LPS or TNF-α; LPS or TNF-α, together with calpeptin; calpeptin alone as a positive control. After treatment, total RNA was extracted, and miR-26b levels were measured by RT-qPCR, as described above.

### 4.18. Other Laboratory Assays

White blood cell (WBC) count and platelet count with MPV were determined by Advia 2120 Hematology System (Bayer Diagnostics, Tarrytown, NJ, USA). Serum C-reactive protein (CRP) and procalcitonin (PCT) levels were measured by electro-chemiluminescent immunoassay using a Cobas 8000 analyzer (Roche Diagnostics, Mannheim, Germany). Soluble P-selectin concentrations were determined in 10 randomly selected plasma samples from each study cohort by commercially available ELISA kit (R&D Systems, Minneapolis, MN, USA) according to the manufacturer’s instructions. Before performing this analysis, samples were thawed and then centrifuged at 10,000 *g* for 1 min.

### 4.19. Data Presentation and Statistical Analyses

Data were expressed in the median with (IQR, interquartile range), or mean ± standard deviation (SD), or standard error of the mean (SEM), as appropriate. A comparison of multiple groups was performed using ANOVA or Kruskal–Wallis with post hoc test, while t-test or Mann–Whitney U test and Chi-squared test were performed to compare two groups of data. The Kolmogorov–Smirnov test was used for the evaluation of the normality of the data. *p* < 0.05 probability level was regarded as statistically significant. Analyses were performed using GraphPad Prism, version 6.01 (GraphPad Software, La Jolla, CA, USA).

## Figures and Tables

**Figure 1 ijms-21-00866-f001:**
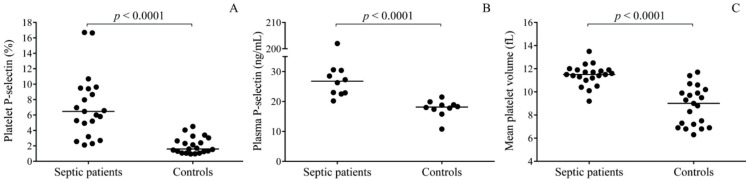
Evaluation of the level of platelet activation status in septic patients (*n* = 21) via surface P-selectin positivity (**A**) and soluble P-selectin levels (**B**) measured in some selected plasma samples (*n* = 10/group) in comparison to controls. Mean platelet volume values were also analyzed to estimate the pool of larger and younger platelets (**C**). All these parameters were significantly higher than controls, suggesting an enhanced level of platelet activation status after the development of sepsis. Dots represent single results, and median values are depicted. Mann-Whitney U test was performed for comparison.

**Figure 2 ijms-21-00866-f002:**
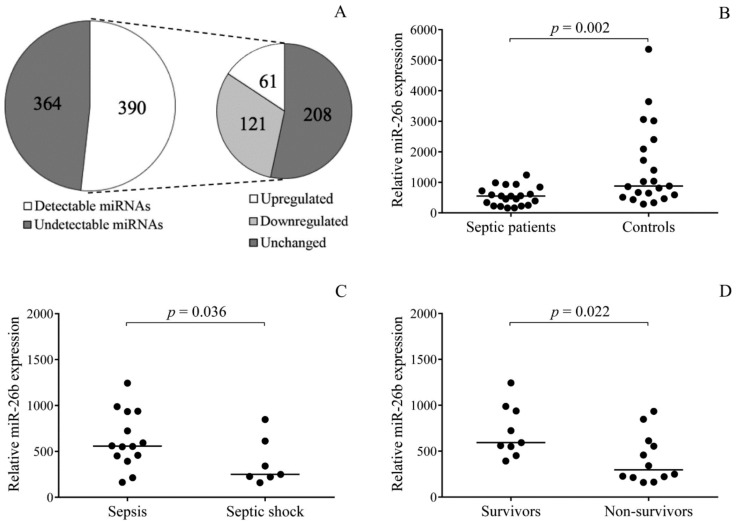
Analysis of abnormal platelet miRNA expression induced by sepsis. First, miRNAs were profiled by TaqMan Open Array in three septic and control samples (**A**). Out of 390 detected, there were 121 significantly downregulated and 61 upregulated miRNAs in septic platelets. Platelet miR-26b (**B**) was validated in both study groups (*n* = 21/group), showing attenuated levels in sepsis. Alterations in miRNA expression were further observed in the septic cohort in relation to disease severity (**C**) and sepsis-related mortality (**D**). Levels of miR-26b reflected septic shock and early death. Dots represent single expression values. Median values are depicted. Mann–Whitney *U* test was performed for comparison.

**Figure 3 ijms-21-00866-f003:**
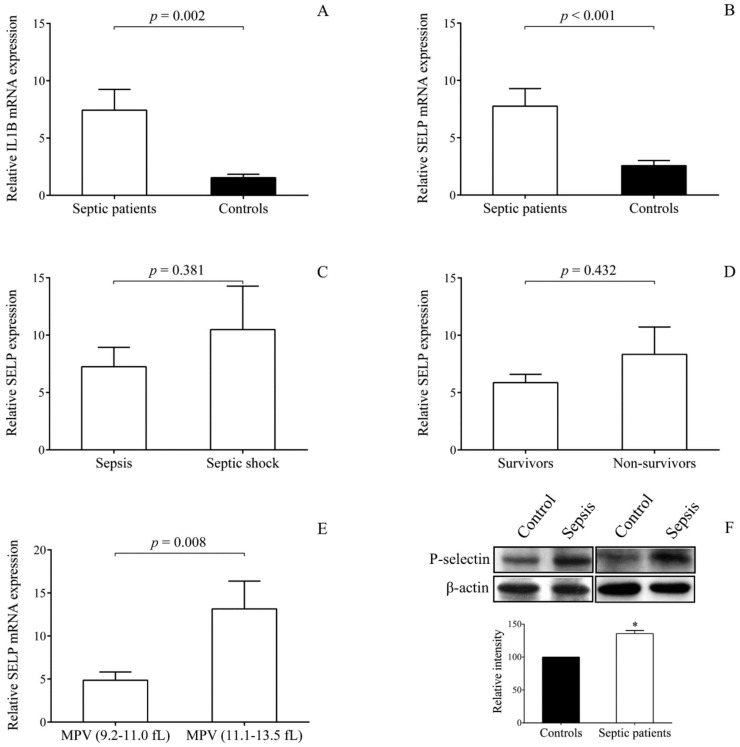
Increased *SELP* and P-selectin protein expression in septic platelets. Platelets of septic patients (*n* = 21) showed elevated levels of *IL1B* (**A**) and *SELP* mRNA (**B**) levels compared to control platelets. There was a tendency for higher platelet *SELP* expression in those sepsis individuals who had septic shock (**C**) or died by sepsis (**D**). In addition, significantly elevated *SELP* mRNA levels were observed in platelets with larger than normal MPV values (*n* = 8) (**E**). Western blot analysis of platelet lysates obtained after 72 h of sepsis onset (*n* = 5) demonstrated increased expression of P-selectin protein compared to normal samples (**F**). Mean ± SEM was depicted, * *p* < 0.05. Mann–Whitney *U* test or unpaired *t*-test was performed for the comparisons.

**Figure 4 ijms-21-00866-f004:**
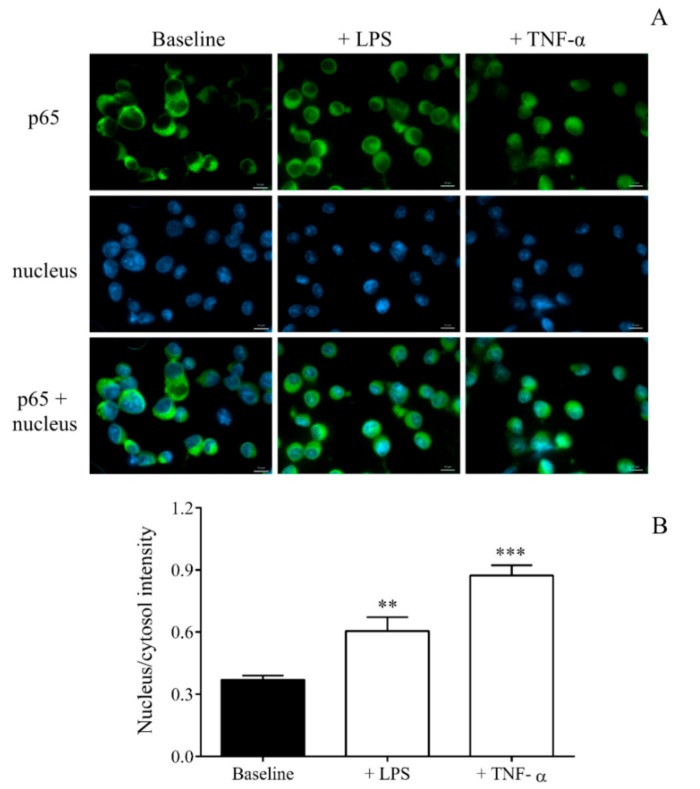
Immunohistochemical staining and analysis of the nuclear factor kappa B (NF-κB) pathway activation in LPS or TNF-α treated MEG-01 cells. Megakaryocytes were stimulated with PBS (baseline), 100 ng/mL LPS or TNF-α for 4 h. Nuclear localization of the NF-κB p65 subunit was monitored by immunostaining. Green: p65 staining; blue: cell nuclei. Scale bar: 20 μm (**A**). The ratio of the fluorescence intensity of the NF-κB immunostaining in cell nuclei and cytosol was analyzed (**B**). Mean ± SEM, *n* = 6–10/group. ** *p* < 0.01 and *** *p* < 0.001 based on statistical analyses.

**Figure 5 ijms-21-00866-f005:**
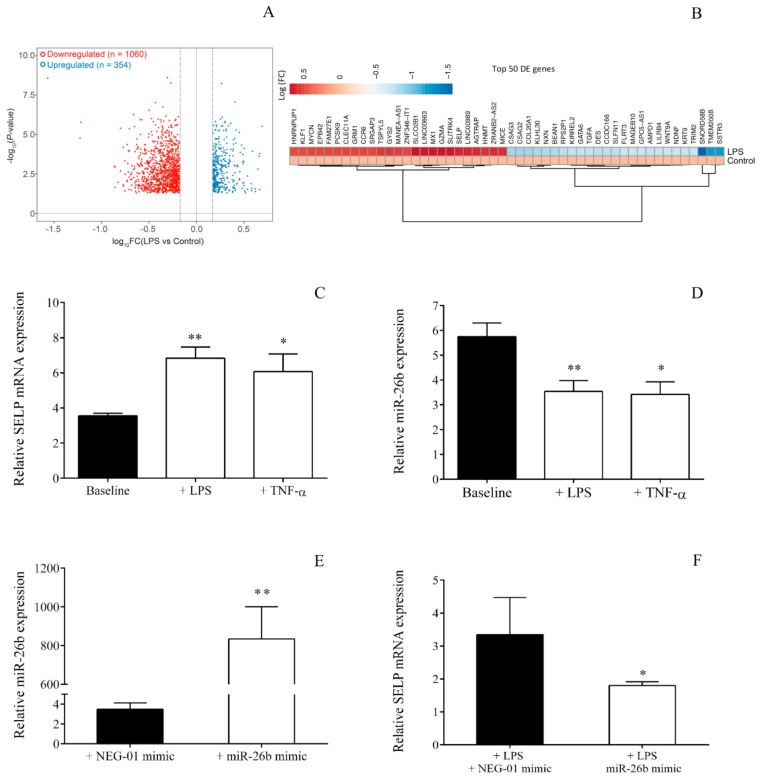
Toll-like receptor (TLR)4 pathway induced changes in mRNA and miRNA expression of MEG-01 cells. Volcano plot depicting the number of differently expressed genes in untreated and lipopolysaccharide (LPS)-treated MEG-01 cells (*n* = 3/condition) at 4 h, observed by RNA-seq analysis, quantified by the DESeq method and plotted based on Log10 (Fold Change) and –log (*p*-value). There were 354 significantly upregulated, and 1060 downregulated transcripts in LPS-activated MEG-01 cells compared to control cells at a fold change (FC) of ≥1.5. Dotted lines indicate FC cut-off (**A**). Clustered heat map of the top 50 differentially expressed genes in LPS-treated MEG-01 cells at 4 h. Genes were displayed as Log (Fold Change) (**B**). We then validated *SELP* expression in MEG-01 cells after LPS treatment (*n* = 6–8/experiment), and TNF-α stimulation was used as a positive control (**C**). In parallel, miR-26b was significantly attenuated by LPS or TNF-α vs. untreated cells (**D**). These results were in accordance with the findings in ex vivo septic platelets. The functional relationship between miR-26b and *SELP* expression was analyzed using specific miRNA mimic transfection in LPS-stimulated MEG-01 cell cultures. The overexpression of miR-26b by mimic (**E**) resulted in lowered SELP mRNA levels compared to samples with control NEG-01 mimic (**F**). Mean ± SEM was depicted, * *p* < 0.05, ** *p* < 0.01 based on statistical analyses.

**Figure 6 ijms-21-00866-f006:**
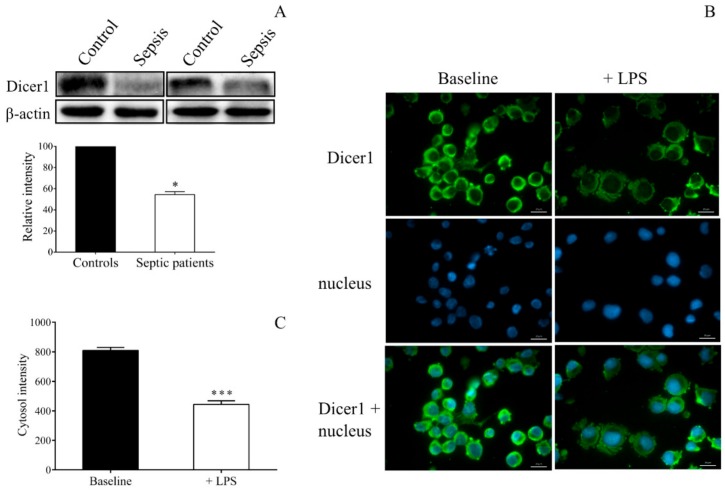
Investigation of Dicer1 level in septic platelets and LPS-stimulated MEG-01 cells in regard to altered miRNAs. First, septic platelets (*n* = 5) within 24 h of the onset of sepsis were studied for the Dicer1 protein level by Western blotting (**A**). We found a significantly lower expression of Dicer1 in septic platelets. LPS-stimulated MEG-01 cell cultures were intracellularly analyzed for Dicer1 positivity with a fluorescence microscope, showing decreased intensity after LPS treatment (**B**). Fluorescence intensity of Dicer1 immunostaining in the cytosol was analyzed (*n* = 15/experiment) (**C**). Green: Dicer1 staining; blue: cell nuclei. Scale bar: 20 μm. Mean ± SEM was depicted, * *p* < 0.05, *** *p* < 0.001 based on statistical analyses.

**Figure 7 ijms-21-00866-f007:**
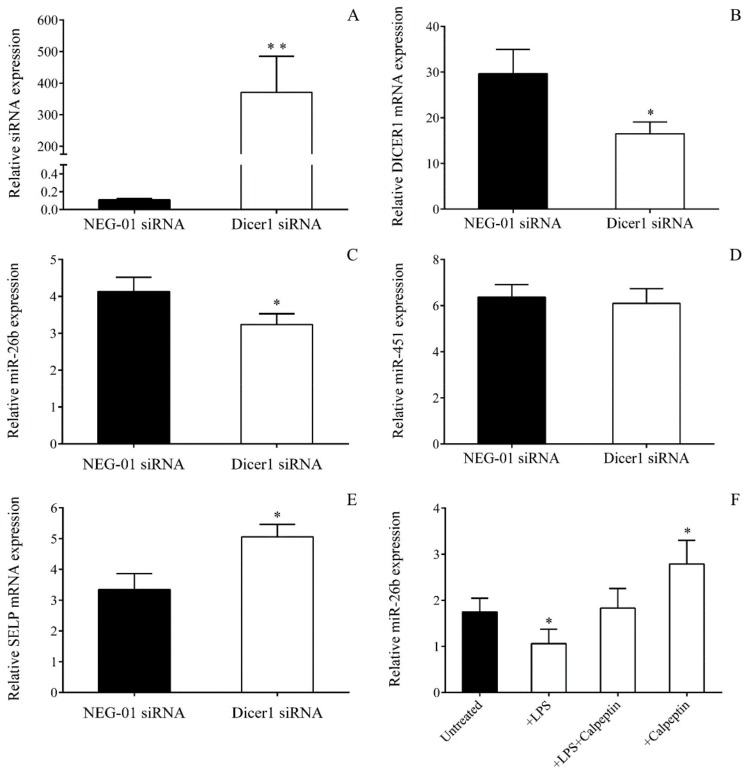
Investigation of the functional role of Dicer enzyme in attenuated miRNA levels in sepsis. First, to mimic sepsis-induced alteration of the Dicer1 level, *DICER1* expression was downregulated via siRNA transfection in MEG-01 cells for the analysis of miR-26b and its target *SELP* mRNA. Dicer1 specific siRNA was successfully transfected into MK cell cultures based on the highly elevated level of Dicer1 siRNA measured by RT-qPCR (**A**). This resulted in decreased *DICER1* mRNA expression compared to control samples with NEG-01 siRNA (**B**). In these cells, reduced miR-26b (**C**) with elevated *SELP* mRNA levels were quantified (**E**). In addition, Dicer1-independent miR-451 (**D**) was analyzed to double-check the specificity of siRNA transfection that showed no change due to manipulation. Second, calpain inhibition was applied to alter miRNA levels in MEG-01 cells among inflammatory conditions *in vitro*. Calpeptin significantly restored miR-26b (**F**) levels when was used during LPS treatment, suggesting the role of abnormal Dicer function in miRNA expression of sepsis. Mean ± SEM was depicted (*n* = 4–6/experiment). * *p* < 0.05, ** *p* < 0.01 vs. NEG-01 or untreated samples based on statistical analyses.

**Table 1 ijms-21-00866-t001:** Summary of demographical and clinical characteristics of septic and control study groups. Data are expressed as median with (IQR, interquartile range), or mean ± SD as appropriate. Sequential organ failure assessment (SOFA) score was applied to determine the extent of organ function or rate of failure. For statistical analysis, we used Student’s t-test or Mann–Whitney U test and Chi-square test as appropriate. WBC: white blood cell; PLT: platelet; CRP: C-reactive protein; PCT: procalcitonin; ICU: intensive care unit; n.s., not significant; n/a, not applicable.

Parameters	Septic Patients (*n* = 21)	Controls (*n* = 21)	*p* Value
Age (years)	64 (51–70)	58 (42–65)	n.s.
Male/female gender (*n*)	16/5	14/7	n.s.
WBC count (G/L)	11.4 (8.3–16.2)	7.6 (6.2–8.9)	*p* < 0.001
PLT count (G/L)	218 (175–264)	332 (290–365)	*p* < 0.01
Serum CRP (mg/L)	210.5 ± 98.2	1.4 ± 1.0	*p* < 0.001
Serum PCT (µg/L)	27.4 ± 11.7	n/a	-
SOFA-score	11 (9–13)	n/a	-
Sepsis/septic shock (*n*)	14/7	n/a	-
ICU length of stay (days)	25.1 ± 14.4	n/a	-
28-day mortality (*n*)	9	n/a	-
Source of infection—pneumonia (*n*)	18	n/a	-
Source of infection—urinary tract (*n*)	3	n/a	-
Organism—Gram-positive bacteria (*n*)	6	n/a	-
Organism—Gram-negative bacteria (*n*)	12	n/a	-
Unknown infection (*n*)	3	n/a	-
Anti-platelet therapy (*n*)	16	12	n.s.

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
