# Peer review of "Reduced miR-26b Expression in Megakaryocytes and Platelets Contributes to Elevated Level of Platelet Activation Status in Sepsis"

_ijms, 2020, doi:10.3390/ijms21030866_

Round 1

Reviewer 1 Report

In this article, the authors came up with a nice idea, though there are some concerns related to this article:

Human subjects are only 21 in this article, which is not sufficient to conclude this kind of title: "Sepsis modulated microRNA expression in megakaryocytes and platelets results in elevated platelet activation". Need more human subject data in both groups. Introduction is not satisfactorily written.  Figure 3D, author should incorporate more images of the western blot bands of P-selectin and beta actin, though its a representative image of the data, but one band can't justify the point.  Figure6A has the same issue like figure 3D.

Reviewer 2 Report

In their article: „Sepsis modulated microRNA expression in megakaryocytes and platelets results in elevated platelet activation“, Szilágyi and colleagues address the influence of microRNA abundance and function in platelets and megakaryocytes during the septic situation. The investigators correlate miRNA expression in sepsis and septic shock diagnosed according to the actual Sepsis-3 criteria, hence under high quality control of patient cohorts. They further focused on an interesting candidate, miR-26b, which targets SELP encoding P-selectin, a highly abundant membrane protein in alpha-granules of platelets. Furthermore, MEG-01 cell line was used as megakaryocyte model to mimic sepsis after LPS treatment. The study is of particular relevance to the field and bears novelty. However, I still see some concerns which need to be addressed:

Please, revise the abstract. At some points performed experiments, conclusion and following analyses are confusing. In the introduction, the authors claim that “septic platelets typically show increased surface P-selectin expression”. Does this mean, that sepsis platelets are pre-activated? The literature is still contradictory regarding this, with several papers showing hyporeactive platelets (e.g. Yaguchi et al., 2004, JTH; Adamzik et al., 2012, Critical Care). I suggest at least to address this paradox. Please, explain to the unexperienced reader which function calpain (calpeptin treatment) has regarding Dicer signalling. The mean of the SOFA score was calculated, which is not informative/correct. Please, use the median here. It would be helpful to indicate the exact time point of patient recruitment (or at least the time frame) throughout the whole manuscript. Figure 1: Why is MPV a parameter for platelet activation, as claimed? It is rather used as indirect marker for increased platelet biogenesis. I am very reluctant regarding this issue. Have further activation markers been used (e.g. GPIIbIIIa binding antibody PAC-1; AnnexinV)? I suggest to rephrase “platelet activation” in “platelet pre-activation”, since activation is measured after agonist treatment by FACS, aggregometry etc. Figure 2B: Could the authors give a statement about the broad range of miR-26b expression in controls? Was this correlated with anti-platelet therapy in those healthy individuals? I think, this is of high relevance, since clopidogrel and aspirin could induce platelet transcriptome and hence miRNA abundance. Figure 2D: Especially when comparing a particular parameter with survival, the time point of patient recruitment and hence measurement is essential to not overstate the correlation. In results section 2.4 the authors claim that “the levels of SELP mRNA was markedly higher in subjects who did not survive”. Please, show the data even though statistical significance is missing. Figure 3C: What is the relevance to correlate SELP expression and MPV? It is not surprising, that larger platelets show higher miRNA abundance. Results section 2.5: The term “experimental sepsis” is highly overstated. I strongly recommend to re-phrase this. In the used in vitro conditions many parameters are missing to really simulate the sepsis situation. Why is soluble CD14 added to the MEG-01 medium? Figures 5C-F: What are the mentioned “MKs”? MEG-01 cells? Be precise. Is this just a methodical confirmation of RNAseq via qPCR? How is the baseline set (delta-delta Ct etc.)? Figure 6B: Please, indicate the markers used for staining according to Figure 4A. Figure 7: Again, how was the baseline/threshold for the qPCR analysis set? The relative expression of controls is not compliant. Figure 7A: How does siRNA measurement indicate transfection efficiency of this particular siRNA? Could it also be siRNA detected in the medium (not intra-cellular)? Figure 7F: It remains unclear if TNF-alpha and LPS were used in combination ore separate. Please, be precise here. What is the overall conclusion of the study in one sentence, since it is not clearly stated? Could the authors provide something like a signalling axis (e.g. sepsis -> lowered Dicer expression/elevated Dicer degradation -> miR-26b down-regulated -> SELP upregulated -> platelet pre-activation)?

Round 2

Reviewer 1 Report

I would like to thank the authors the address all the queries. And will accept in present form.